# Acetylsalicylic Acid Supplementation Affects the Neurochemical Phenotyping of Porcine Duodenal Neurons

**DOI:** 10.3390/ijms24129871

**Published:** 2023-06-07

**Authors:** Marta Brzozowska, Jarosław Całka

**Affiliations:** Department of Clinical Physiology, Faculty of Veterinary Medicine, University of Warmia and Mazury in Olsztyn, Oczapowskiego Str. 13, 10-718 Olsztyn, Poland; calkaj@uwm.edu.pl

**Keywords:** acetylsalicylic acid, ENS, peripheral nervous system, duodenum, pig

## Abstract

Aspirin (ASA) is a popular nonsteroidal anti-inflammatory drug (NSAID), which exerts its therapeutic properties through the inhibition of cyclooxygenase (COX) isoform 2 (COX-2), while the inhibition of COX-1 by ASA results in the formation of gastrointestinal side effects. Due to the fact that the enteric nervous system (ENS) is involved in the regulation of digestive functions both in physiological and pathological states, the aim of this study was to determine the influence of ASA on the neurochemical profile of enteric neurons in the porcine duodenum. Our research, conducted using the double immunofluorescence technique, proved an increase in the expression of selected enteric neurotransmitters in the duodenum as a result of ASA treatment. The mechanisms of the visualized changes are not entirely clear but are probably related to the enteric adaptation to inflammatory conditions resulting from aspirin supplementation. A detailed understanding of the role of the ENS in the development of drug-induced inflammation will contribute to the establishment of new strategies for the treatment of NSAID-induced lesions.

## 1. Introduction

Aspirin (acetylsalicylic acid; ASA) belongs to the group of nonsteroidal anti-inflammatory drugs (NSAIDs) and is considered one of the most commonly used drugs. It is well known that side effects linked to NSAID treatments are mainly associated with the gastrointestinal tract (GIT) [1]. Changes in the upper GIT may be unimportant (ecchymosis, erosions) or significant (ulcers, oesophagitis). About half of the patients taking NSAIDs regularly report gastric erosions, and 15% to 30% complain of gastric ulcers [2]. The most serious negative effects are bleeding as well as perforations that NSAIDs can cause in both ulcerative and non-ulcerative states throughout the gastrointestinal tract. Interestingly, major changes can occur without any prior symptoms [1,2].

ASA was originally used as an anti-inflammatory drug. Due to the fact that ASA is an irreversible inhibitor of cyclooxygenase (COX), which produces prostaglandins (PGs) and thromboxane precursors, it has also found several other applications, often unrelated to its original purpose [3]. It is well known that acetylation of platelet COX-1 at serine-529 is the direct mechanism of action of low-dose (75–300 mg) aspirin as an antiplatelet agent [1,4]. In turn, ASA used in higher doses (above 300 mg) causes the inhibition of PG synthesis by blocking COX-1 and COX-2. Through this mechanism, ASA exhibits anti-inflammatory and antipyretic effects [4].

It should also be mentioned that ASA inhibits COX irreversibly by acetylation, unlike other NSAIDs [5]. When COX is irreversibly blocked, PG production returns to normal levels after a few days when a new enzyme is synthesized. In the stomach, complete recovery of PG levels was noted after 8 days when low daily doses were used [5]. This finding supports the statement that aspirin is one of the most important blockers of PG synthesis.

Due to the high therapeutic application of aspirin, a thorough understanding of the mechanism of drug action and the process of the formation of inflammatory changes is necessary. Undoubtedly, the enteric neurons are involved in the gastrointestinal response to external factors.

The innervation of the GIT consists of several populations of neurons whose cell bodies are located outside (extrinsic) or in the wall (intrinsic) of the digestive tract. The functioning of the intestines is locally controlled by the enteric nervous system (ENS), organized from three intramural plexuses—the myenteric plexus (MP), the inner submucosal plexus (ISP) and the outer submucosal plexus (OSP) [6]. The proper functioning of the organ depends on the precision of intercellular signaling between the enteric neural network and epithelial and immune cells that are prone to disorders associated with the development of inflammation. Nerve cells react to irritants by changing the expression of selected neurotransmitters regulating gut activity [7,8].

It should be noted that each neuron synthesizes different neurotransmitters, which create its neurochemical code. Immunohistochemical methods revealed the presence of about 50 different neuroactive substances [9]. Galanin (GAL) is a neuropeptide found in intramural neurons in the digestive tract and regulates secretion and contractility [10]. The cocaine- and amphetamine-regulated transcript peptide (CART) is widely distributed in both the central and peripheral nervous systems; however, the exact function of CART in gastrointestinal physiology is poorly understood. Previous research has revealed that CART can regulate gastric acid secretion and colon motility [11]. In the GIT, the pituitary adenylate cyclase-activating polypeptide (PACAP) is responsible for regulating the synthesis of hormones, stimulating the secretion of digestive juices, affecting smooth muscle contractility cell migration and proliferation and local blood flow [12]. In turn, nitric oxide (NO) synthesized by neural nitric oxide synthase enzyme (nNOS) is an inhibitory neurotransmitter of the GIT [13]. Another neurotransmitter, substance P (SP), has been identified as an important mediator in the development and progression of intestinal inflammation [12]. The vasoactive intestinal polypeptide (VIP) is an immunomodulatory agent that has proven anti-inflammatory properties. VIP has been reported to inhibit the release of pro-inflammatory cytokines and mediators such as interleukin (IL) -6, IL-12, tumor necrosis factor alpha (TNF-α) and chemokines [14].

It should also be mentioned that the anatomical organization, as well as the neurochemical characteristics of intrinsic neural networks responsible for the functioning of intestinal processes, appear to be more complex in larger mammals, including humans, than in small laboratory animals [15]. Due to the structural and functional similarities, the porcine intestine is considered a suitable experimental model for studying the physiology of the human ENS, as well as its plasticity and response to an external stimulus.

The duodenum, especially in its upper portion, is particularly exposed to irritating substances such as gastric acid, pepsin, as well as other damaging factors, e.g., NSAIDs, which enter the intestines as a result of gastric emptying [16,17]. It should be noted that the epithelium of the duodenal mucosa is less tight compared to the gastric epithelium. On the other hand, the protective mucus layer is much thicker, which is the first line of duodenal defense [18].

Determining the chemical coding of porcine duodenum, which is particularly vulnerable to the adverse effects of acetylsalicylic acid [19], can determine the potential involvement of enteric neurotransmitters in the development of inflammatory lesions. At the same time, an analysis of the obtained results may deepen the knowledge about the potential use of neuroactive substances in the therapy of the negative effects of aspirin treatment. Accordingly, the purpose of this experiment is to determine the effect of acetylsalicylic acid (in high doses, used in inflammation and fever therapy) on the neurochemical characterization of enteric neurons in the porcine duodenum.

## 2. Results

Post-mortem analysis of collected materials showed pathological signs in experimental pigs. On the duodenal mucosa, inflammatory changes, such as ecchymosis, hyperemia, edema and excessive mucus production, were observed in animals that received aspirin.

The immunoreactivity to all tested neurotransmitters was observed in all enteric plexuses within the porcine duodenum in both the study and control pigs. Aspirin supplementation caused an increase in the number of CART-, PACAP-, SP-, VIP-, nNOS- and GAL-immunopositive neurons (Figure 1).

The vasoactive intestinal polypeptide is a neuroactive substance commonly found in the enteric nervous system. In the physiological state within the myenteric plexuses, the expression of this messenger was estimated at 11.90 ± 0.30% (Figure 2(A1–A3)), while aspirin treatment caused an increase in the number of VIP-IR neurons to 13.79 ± 0.98% (Figure 2(B1–B3)). The degree of VIP expression in the OSP in the control group was 13.78 ± 1.07% (Figure 2(C1–C3)), while aspirin supplementation caused an increase in the percentage of the neurotransmitter to 20.64 ± 0.53% (Figure 2(D1–D3)). In inner submucosal plexuses, 17.63 ± 0.77% of neurons in the control group showed neuronal immunoreactivity to VIP (Figure 2(E1–E3)), while 21.80 ± 1.32% of VIP-positive neurons were noted in the experimental group (Figure 2(F1–F3)).

Up to 18.50 ± 0.26% of nNOS-positive neurons in the control group were observed within the myenteric cells (Figure 3(A1–A3)). Aspirin administration increased nitric oxide expression in MP to 20.22 ± 0.60% (Figure 3(B1–B3)). Control animals had 8.40 ± 0.53% nNOS expression in OSP (Figure 3(C1–C3)), while treatment with acetylsalicylic acid resulted in an increase in nNOS immunoreactivity (up to 11.17 ± 0.90%) (Figure 3(D1–D3)). A similar relationship was found in the inner submucosal plexuses. In the physiological state, 6.39 ± 0.73% of nNOS positive nerve cells were counted (Figure 3(E1–E3)) and, as a result of aspirin administration, the percentage increased to 10.41 ± 0.43% (Figure 3(F1–F3)).

An analysis of the results showed that in control animals, 8.74 ± 0.63% of neurons revealed expression against substance P in the myenteric cells (Figure 4(A1–A3)). Aspirin treatment increased the number of SP-IR neurons to 14.98 ± 0.49% (Figure 4(B1–B3)). In the control group, 20.22 ± 1.38% of neurons were expressed relative to SP in the OSP (Figure 4(C1–C3)). An increase in SP-positive neurons (up to 25.14 ± 1.43%) was observed in experimental animals (Figure 4(D1–D3)). In the ISP, aspirin caused an increase in the percentage of neurons showing the presence of substance P to 22.56 ± 0.33% (Figure 4(F1–F3)). In the control pigs, the number of SP-IR cells was estimated at 20.80 ± 0.25% (Figure 4(E1–E3)).

Galanin was another substance tested during the described experiment. The neurotransmitter has been previously reported within the myenteric plexuses. In a physiological state, it was estimated that 1.36 ± 0.12% of neurons exhibit GAL immunoreactivity (Figure 5(A1–A3)). Aspirin treatment resulted in an increase in GAL-IR neurons by two percentage points (up to 3.53 ± 0.20%) (Figure 5(B1–B3)). In the outer submucosal plexuses, the increase in galanin expression was significantly noticeable. In the control group, 19.71 ± 1.48% of GAL-positive neurons were observed (Figure 5(C1–C3)), while supplementation with acetylsalicylic acid caused an increase in galanin expression to 26.71 ± 0.35% (Figure 5(D1–D3)). In ISP, experimental animals had increased galanin expression (35.46 ± 1.47%) (Figure 5(F1–F3)) compared to the control group (29.67 ± 2.02%) (Figure 5(E1–E3)).

In turn, the expression of CART peptide in a physiological state was 13.61 ± 0.38% in the neurons collected into myenteric plexuses (Figure 6(A1–A3)). Aspirin treatment induced an increase in CART immunoreactivity (15.42 ± 0.25%) in MP (Figure 6(B1–B3)). The same dependencies were noted in the submucosal plexuses. 15.88 ± 0.15% of neurons of the outer submucosal plexuses showed CART immunoreactivity in the control group (Figure 6(C1–C3)), while aspirin supplementation increased these neurons to 20.25 ± 0.51% (Figure 6(D1–D3)). In ISP, 10.55 ± 0.14% CART-IR neurons were observed in control gilts (Figure 6(E1–E3)). In animals receiving acetylsalicylic acid, the percentage of CART-positive nerve cells was 13.54 ± 1.16% (Figure 6(F1–F3)).

Pituitary adenylate cyclase-activating polypeptide (PACAP) is widely noted in all parts of the enteric nervous system. In the myenteric plexus neurons, the PACAP peptide expression in animals receiving empty gelatin capsules was 9.69 ± 0.37% (Figure 7(A1–A3)), whereas the animals in the study group showed an increase in PACAP-positive neurons (to 14.02 ± 0.17%) (Figure 7(B1–B3)). In OSP in the control animals, the presence of PACAP was noted in 10.51 ± 0.31% of PGP 9.5-like immunoreactive cells (Figure 7(C1–C3)). Aspirin treatment induced an increase in PACAP-IR nerve cells to 15.61 ± 0.59% (Figure 7(D1–D3)). In turn, within the inner submucosal plexuses, an increase in PACAP expression by 5 pp was observed in the study group (17.45 ± 0.88%) (Figure 7(F1–F3)) compared to the control animals (12.82 ± 0.35%) (Figure 7(E1–E3)).

## 3. Discussion

NSAIDs have gained popularity because of their effectiveness in treating fever, pain and inflammatory conditions [1]. NSAIDs, beneficial for their anti-inflammatory and analgesic effects, account for 8% of prescriptions worldwide. Furthermore, there has been an increase in over-the-counter use, with 26% using more than the recommended dose [1]. Their effectiveness is limited due to the side effects affecting the stomach and intestines.

The results of this study revealed that the regular and long-term administration of aspirin caused inflammatory changes in the porcine duodenum. The observed phenotypic changes in duodenal enteric neurons may be a response of neurons to the harmful effects of ASA. The marked upregulation of CART, PACAP, GAL, VIP, nNOS and SP revealed in this study suggests the involvement of intestinal neurotransmitters in the mechanism of formation of aspirin-induced duodenal lesions. These results are in line with our previous reports [7,20,21].

It is well known that NSAIDs exert their therapeutic effects by inhibiting COX. There are two isoforms of the COX enzyme: COX-1 and COX-2 [22,23]. The COX-1 isoform is expressed in most tissues and produces PGs involved in the protective mechanisms of the gastrointestinal tract and increased mucus production. Thus, COX-1 blockade by NSAIDs creates a vulnerable environment in the digestive system. COX-2 is an induced isoform whose production is stimulated by inflammation. Its effects are mediated by prostaglandins involved in inflammatory processes as well as fever and pain [23].

As mentioned above, PGs have a protective function in the GIT by stimulating mucus production. It has also been proven that these eicosanoids are involved in gastrointestinal functioning, including secretion and motility [24]. Decreased PG synthesis due to aspirin administration may have resulted in decreased duodenal secretion. The demonstrated overexpression of neurotransmitters with known secretolytic (CART) or motility properties (such as VIP or PACAP) may be a response of enteric neurons to the disturbance of homeostasis within the duodenum [25].

It is highly probable that enteric neurotransmitters may be involved in aspirin-induced inflammatory changes in the duodenum caused by aspirin COX inhibition. This is evidenced by the fact that the increase in the expression of the studied neurosubstances was visible in gastrointestinal inflammatory conditions. In patients with Helicobacter pylori infection, increased expression of VIP was found in the myenteric neurons in the stomach [26]. Similar results were observed in the rat diabetes model. In the myenteric plexus of the ileum, diabetes caused a threefold increase in VIP expression [27]. Nitric oxide produced within the gastrointestinal tract is largely responsible for relaxing the smooth muscle coat and regulating the lumen of blood vessels [13]. Decreasing the lumen may have an adverse effect on the motor processes and the blood flow and thus the resorption of nutrients, which is particularly important within the duodenal area. CART expression has been described throughout the gastrointestinal tract in many animal species as well as in humans [11]. The function of this peptide in the GIT is mainly related to the regulation of motor processes and the passage of the food content. Therefore, the increase in the number of CART-positive neurons is evidence of the adverse effects of aspirin on the gastrointestinal tract.

Another proposed mechanism of NSAID-induced gastrointestinal injury involves the stimulation of neutrophil adherence to the vascular endothelium in the gastrointestinal microcirculation. This is confirmed by the fact that in neutropenic rats, no gastrointestinal damage was observed after the administration of NSAIDs. It is tempting to speculate that the observed increase in PACAP expression may be related to the increased recruitment of neutrophils caused by drug-induced duodenal injury. PACAP has been suggested to have a pleiotropic effect and is considered an immune modulator [28]. Studies to determine the therapeutic properties of PACAP in murine Toxoplasma-gondii-induced acute ileitis have shown that the numbers of recruited neutrophils, monocytes and macrophages were reduced in PACAP-treated mice [28].

It has also been proven that SP is involved in inflammatory processes in the digestive system and reveals immunomodulatory properties. SP is responsible for the activation of the phosphorylation of protein kinase C δ as well as the intensification of the synthesis of pro-inflammatory cytokines such as IL-8, IL-6, TNF-α and IL-1β [29]. Due to the fact that SP may play a key role in the development of colitis, it seemed reasonable to investigate its expression in the course of aspirin-induced duodenal inflammatory lesions [30].

Many studies have shown that SP expression in the GIT was increased as a result of inflammation [16,31]. This corresponds to the results obtained in the current experiment, in which the percentage of SP-positive neurons increased in aspirin-treatment pigs. On the other hand, a decrease in the SP level was observed after oral administration of ulcerative compounds, such as cysteamine, dulcerozine or mepirizole, in the inflammatory duodenum [32]. Thus, it can be assumed that the acute phase of inflammation activates the enteric synthesis of SP, while the decrease in the expression of this neurotransmitter is associated with the recovery process.

The local damage to the gastrointestinal mucosa is related to the acidic properties of NSAIDs, which irritate the epithelium [33]. In addition, these drugs contribute to the reduction in cell proliferation and, consequently, increase intestinal permeability. This leads to the development of inflammation and the recruitment of pro-inflammatory mediators such as TNF-α and ILs [34,35]. It is very likely that the local action of aspirin caused an increase in the expression of neurotransmitters in the present study. An interesting hypothesis is, therefore, that the increase in the production of pro-inflammatory factors caused by the ASA administration induced an increase in the expression of protective neurotransmitters in the porcine duodenum. One of the most notorious examples is TNF-α, whose elevated level is found in damaged colon tissue. Galanin supplementation caused a reduction in TNF-α levels [36,37]. McDonald et al. came to similar conclusions indicating that galanin can be highly effective in the treatment of bone diseases in which bone formation may be inhibited due to excessive production of TNFα and IL-1β [38]. In view of the fact that CART expression changes in a pathological state, it is widely accepted that the neuropeptide is also involved in the neuroprotection of enteric cells as a result of ongoing inflammation [7,17]. This finding is supported by a study by Ekblad et al. indicating that CART promotes the survival of ENS neurons in rodents [39].

Somasundaram et al. showed that the local action of NSAIDs causes an increase in the level of NOS, which in turn contributes to the development of erosions and ulcers [40]. It is possible that the increase in nNOS expression observed in this study was the result of the local action of aspirin. In addition, NO, like GAL and CART, has a protective effect on enteric neurons. Nitric oxide exhibits anti-adhesive properties and has been proven to inhibit neutrophil recruitment in the inflammatory area. The anti-adhesive effect also involves inhibiting the production of pro-inflammatory molecules as well as the expression of adhesive compounds [41]. The use of NO synthase inhibitors in trinitrobenzenesulfonic acid (TNBS)-induced ileitis in guinea pigs had positive effects on reducing inflammation. TNBS administration caused an increase in myeloperoxidase levels as well as increased colon wall thickness. Treatment with an NO synthase inhibitor prevented these changes [42].

The neurotransmitters, the expression of which was revealed in the presented study, are involved in gastrointestinal inflammatory processes, as well as regulate the functioning of the digestive system in a physiological state. It should be noted that the exact role of enteric neurosubstances depends on the segment of the GIT as well as on the animal species studied. In the stomach, CART shows a relaxant effect and inhibits gastric emptying, and, within the colon, stimulates intestinal motility [43]. In turn, GAL modifies intestinal motility by increasing or decreasing the release of neuroactive substances and may also act directly by activating receptors located at the smooth muscle cells [10]. The exact function of selected neurosubstances in the digestive system is still the subject of a growing number of studies.

The ENS is composed of two main cell types—neurons and enteric glial cells (EGCs). Various studies have underlined the involvement of EGCs in intestinal inflammation, during which both mucosal and motor functions are altered [44]. In fact, EGCs can synthesize cytokines, and inflammatory conditions modulate glia proliferation [45]. Furthermore, the ablation of EGCs in two transgenic mice models caused fatal intestinal inflammation. Aube et al. concluded that EGCs can induce changes in the chemical coding of enteric neurons [46]. It cannot be ruled out that this relationship also occurred in the presented study; however, this hypothesis needs to be tested.

## 4. Materials and Methods

Immature gilts of the Pietrain × Duroc race weighing about 20 kg were obtained from a breeding farm in Lubawa (Poland) and were used in the present study. The animals were housed under normal laboratory conditions at 20–22 °C and a 12 h dark/light cycle. The pigs were kept in cages for four animals each and were fed twice a day with commercial compound feed adapted to their needs and age. The pigs received humane care, and the experiment strictly followed the ethical guidelines and approval of the Local Ethical Committee in Olsztyn (decision no. 54/2017 from 25 July 2017).

The animals were allowed to adapt to the new environment for seven days prior to the experimental study. The animals were then weighed and randomly divided into two experimental groups—control (C group) and experimental (ASA group) with four gilts per group. The control animals received empty gelatin capsules, while the pigs in the study group were given acetylsalicylic acid (Aspirin; Bayer Bitterfeld GmbH; Bitterfeld-Wolfen, Germany) orally in a dose of 100 mg/kg b.w., once daily approx. one hour before feeding. The animals were weighed once a week to determine the aspirin doses.

After an administration period of 28 days for aspirin, the animals were pretreated with azaperone (Stresnil; Janssen Pharmaceutica N.V., Belgium, 2 mg/kg b.w., i.m.), and 30 min later, the main anesthetic was administered. All animals were killed by overdoses of sodium thiopental (Thiopental, Sandoz, Kundl-Rakúsko, Austria) which was administered intravenously by slow, fractionated infusion. The depth of anesthesia was controlled by monitoring the corneal reflex.

After the cessation of vital functions, tissues (3 cm duodenal fragments approximately 10 cm from the pylorus) were collected for further examination. In the next step, the material was kept in a solution of 4% paraformaldehyde (pH 7.4) for one hour. Duodenal fragments were subsequently washed twice in 0.1 M phosphate buffer (pH 7.4, 4 °C) for three days and then stored in 18% sucrose at 4 °C until being sectioned. Frozen samples were cut into 14 µm sections (about 30–50 scraps per animal) using a cryostat (Microm HM 560 cryostat, Carl Zeiss, Jena, Germany) and affixed to chrome-alum-coated slides.

In the next step of this study, designed preparations were dried at room temperature for 45 min. The PBS solution consisting of (composition in mM) 137 NaCl, 2.7 KCl, 10 NaH_2_PO_4_ and 1.8 KH_2_PO_4_ was then used to rinse the tissues three times with an interval of 15 min. Duodenal sections were placed for 1 h in a “blocking mixture” composed of 10% horse serum and 0.1% bovine serum albumin in 0.1 M PBS, 1% Triton X-100, 0.05% Thimerosal and 0.01% sodium azide. After triple-rinsing in PBS, all sections were immunolabeled (as described previously by Brzozowska et al. [12]). in humid chambers with a mixture of two primary antisera (overnight). The protein gene-product 9.5 (PGP 9.5) was used as a pan-neuronal marker to visualize nerve cells. As primary antibodies, vasoactive intestinal polypeptide (VIP), galanin (GAL), substance P (SP), pituitary adenylate cyclase-activating polypeptide (PACAP), neuronal nitric oxide synthase (nNOS), and cocaine- and amphetamine-regulated transcript peptide (CART) were also used.

On the following day, duodenal sections were washed again in a PBS solution (3 × 15 min) and placed in a mixture of secondary antibodies—Alexa Fluor 488 and/or 546 for 1 h. Table 1 lists the primary and secondary antibodies used in the experiment. After washing (PBS, 3 × 5 min), preparations were covered with a polyethylene glycol solution and subjected to further analysis.

The standard controls, i.e., pre-absorption for the neuropeptide antisera with appropriate antigen: PGP 9.5 (SRP5149, Sigma, St. Louis, MO, USA); GAL (026-06, Phoenix Pharmaceutical); nNOS (N3033, Sigma, St. Louis, MO, USA); SP (05-23-0600, Sigma, St. Louis, MO, USA); VIP (064-24, Phoenix Pharmaceutical); PACAP (A1439, Sigma, St. Louis, MO, USA) and CART (Phoenix Pharmaceuticals, Burlingame, CA, USA) at a concentration of 20 µg/mL for 18 h at 37 °C and the omission, as well as the replacement of all primary antisera by non-immune sera, were performed to test immunohistochemical labeling. The above-mentioned controls completely eliminated labeling in the tissue.

The neurochemical profile of enteric neurons was examined using a fluorescence microscope (Olympus BX51) equipped with epifluorescence and appropriate filter sets. Micrographs were taken using a digital monochromatic camera (Olympus XM 10). The microscope was also equipped with cellSens Dimension Image Processing software version 4.1. To specify the percentage of immunoreactive neurons relative to VIP, nNOS, CART, PACAP, GAL and SP in the duodenum from each type of plexuses (MP, OSP and ISP), the number of immunoreactive neurons versus the tested neurotransmitters was expressed as a percentage of the total number of PGP 9.5-positive cells. At least 500 PGP 9.5 positive neurons per animal were considered in this study. Only neurons with a visible cell nucleus were analyzed. Preparations that were stained with the same antibody mixture were spaced apart by a minimum of 100 µm from each other to prevent analysis of the same cells. Test results were described as mean ± standard error. A *t*-test by Statistica 12 software (StatSoft Inc., Tulsa, OK, USA) was used to determine the significance of differences between study groups. A *p*-value of 0.05 or less was declared statistically significant.

## 5. Conclusions

Long-term administration of aspirin induced a significant increase in VIP-, SP-, nNOS-, GAL-, SP- and PACAP-LI neurons in the porcine duodenum. The observed changes may be caused by the direct influence of NSAIDs on enteric neurons or the inflammation accompanying the treatment. The registered increase in the expression of selected neurotransmitters may be related to the participation of neuropeptides in the regulatory processes of gastrointestinal inflammation. As the negative effects of taking NSAIDs cause damage to the digestive system, the participation of neurotransmitters in the immune response may be used in the future as a tool in the treatment of digestive disorders.

## Figures and Tables

**Figure 1 ijms-24-09871-f001:**
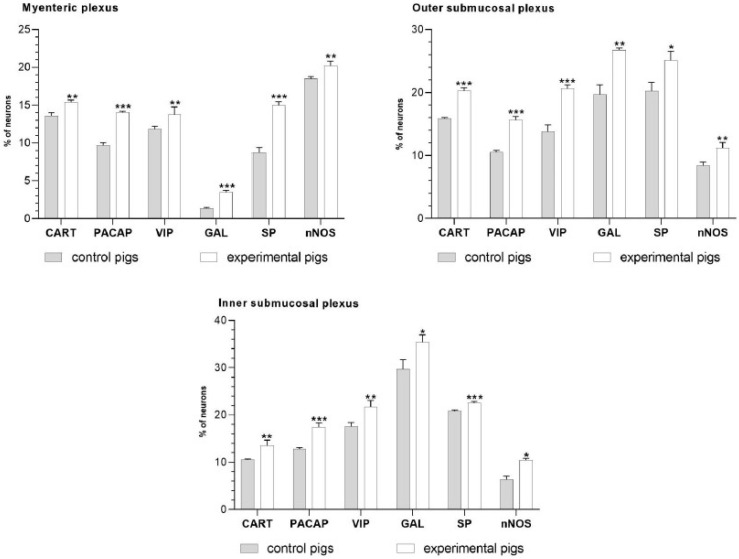
The graphs show the percentage of CART-, PACAP-, VIP-, GAL-, SP- and nNOS-immunoreactive neurons in porcine duodenal enteric plexuses in the control and experimental groups. * *p* < 0.05, ** *p* < 0.01, *** *p* < 0.001 indicates differences in the expression of a particular studied substance in comparison to the control pigs.

**Figure 2 ijms-24-09871-f002:**
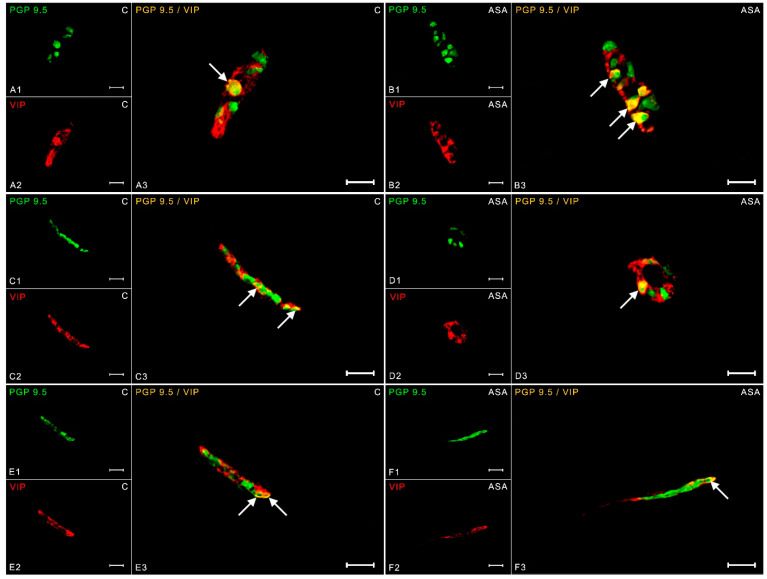
Neuron bodies within myenteric (**A1**–**A3**;**B1**–**B3**), inner (**C1**–**C3**;**D1**–**D3**), and outer (**E1**–**E3**;**F1**–**F3**) submucosal plexuses of porcine duodenum containing the protein gene-product 9.5 (PGP 9.5) (used as a pan-neuronal marker) and vasoactive intestinal peptide (VIP) under physiological conditions (C) and after aspirin supplementation (ASA). The pictures (**A3**–**F3**) have been created by digital superimposition of two color channels, and colocalizations of both antigens in the studied neuron bodies were indicated with arrows. Scale bar size: 50 µm.

**Figure 3 ijms-24-09871-f003:**
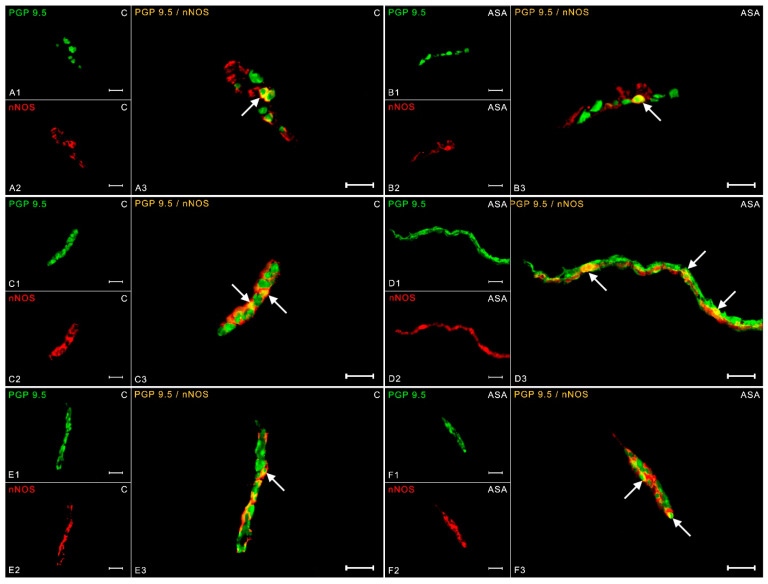
Neuron bodies within myenteric (**A1**–**A3**;**B1**–**B3**), inner (**C1**–**C3**;**D1**–**D3**), and outer (**E1**–**E3**;**F1**–**F3**) submucosal plexuses of porcine duodenum containing the protein gene-product 9.5 (PGP 9.5) (used as a pan-neuronal marker) and neuronal nitric oxide synthase (nNOS) under physiological conditions (C) and after aspirin supplementation (ASA). The pictures (**A3**–**F3**) have been created by digital superimposition of two color channels, and colocalizations of both antigens in the studied neuron bodies were indicated with arrows. Scale bar size: 50 µm.

**Figure 4 ijms-24-09871-f004:**
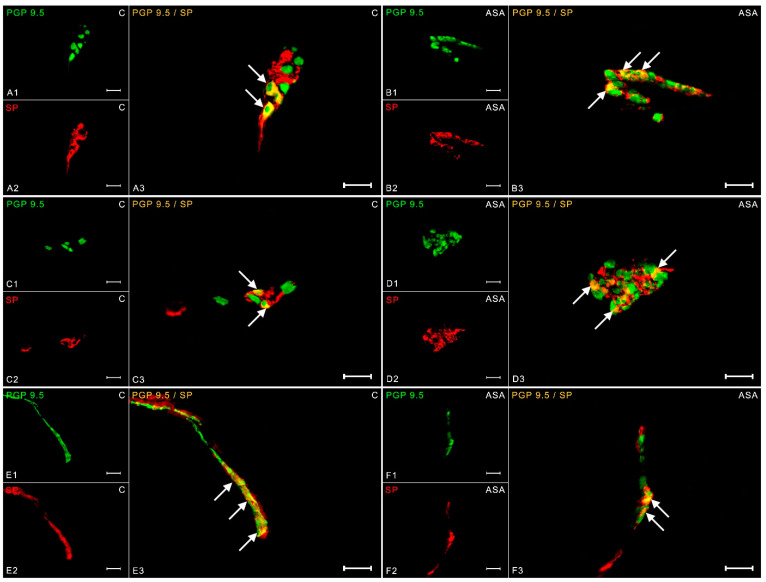
Neuron bodies within myenteric (**A1**–**A3**;**B1**–**B3**), inner (**C1**–**C3**;**D1**–**D3**), and outer (**E1**–**E3**;**F1**–**F3**) submucosal plexuses of porcine duodenum containing the protein gene-product 9.5 (PGP 9.5) (used as a pan-neuronal marker) and substance P (SP) under physiological conditions (C) and after aspirin supplementation (ASA). The pictures (**A3**–**F3**) have been created by digital superimposition of two color channels, and colocalizations of both antigens in the studied neuron bodies were indicated with arrows. Scale bar size: 50 µm.

**Figure 5 ijms-24-09871-f005:**
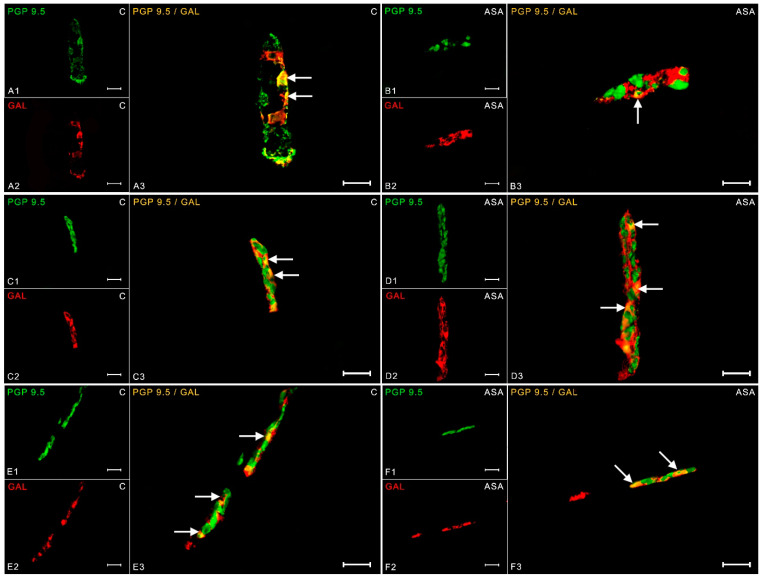
Neuron bodies within myenteric (**A1**–**A3**;**B1**–**B3**), inner (**C1**–**C3**;**D1**–**D3**), and outer (**E1**–**E3**;**F1**–**F3**) submucosal plexuses of porcine duodenum containing the protein gene-product 9.5 (PGP 9.5) (used as a pan-neuronal marker) and galanin (GAL) under physiological conditions (C) and after aspirin supplementation (ASA). The pictures (**A3**–**F3**) have been created by digital superimposition of two color channels, and colocalizations of both antigens in the studied neuron bodies were indicated with arrows. Scale bar size: 50 µm.

**Figure 6 ijms-24-09871-f006:**
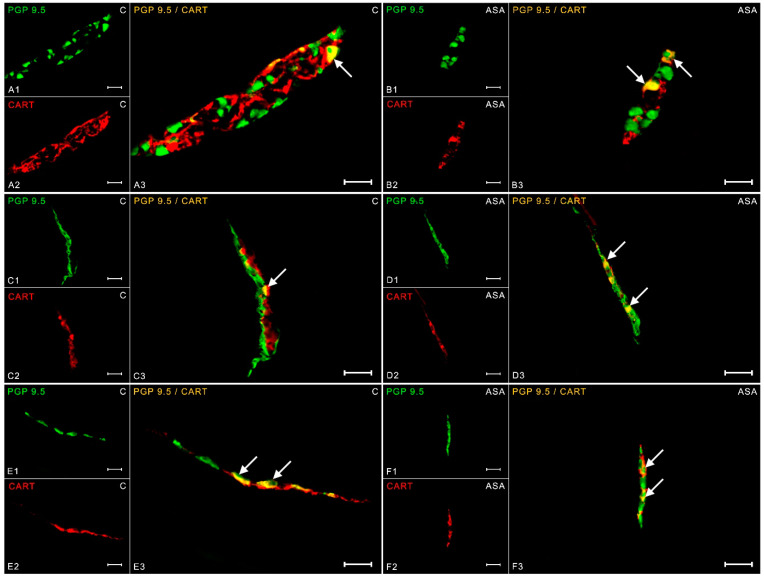
Neuron bodies within myenteric (**A1**–**A3**;**B1**–**B3**), inner (**C1**–**C3**;**D1**–**D3**), and outer (**E1**–**E3**;**F1**–**F3**) submucosal plexuses of porcine duodenum containing the protein gene-product 9.5 (PGP 9.5) (used as a pan-neuronal marker) and cocaine- and amphetamine-regulated transcript peptide (CART) under physiological conditions (C) and after aspirin supplementation (ASA). The pictures (**A3**–**F3**) have been created by digital superimposition of two color channels, and colocalizations of both antigens in the studied neuron bodies were indicated with arrows. Scale bar size: 50 µm.

**Figure 7 ijms-24-09871-f007:**
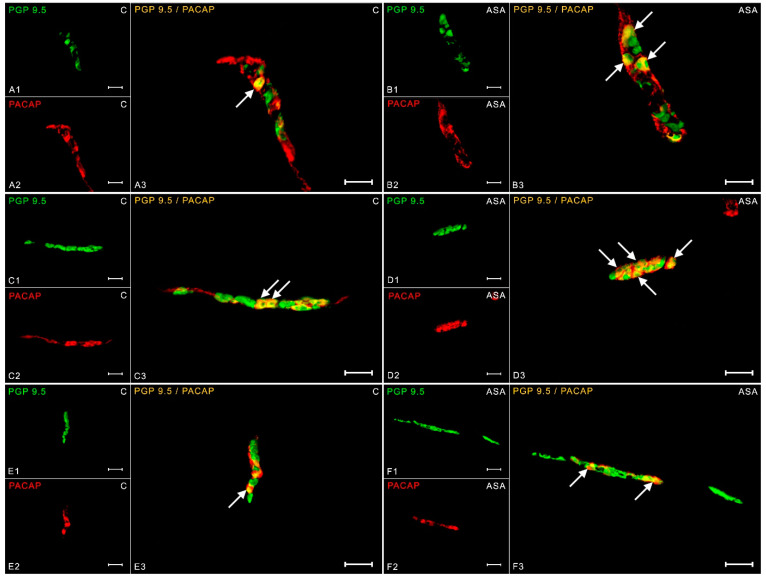
Neuron bodies within myenteric (**A1**–**A3**;**B1**–**B3**), inner (**C1**–**C3**;**D1**–**D3**), and outer (**E1**–**E3**;**F1**–**F3**) submucosal plexuses of porcine duodenum containing the protein gene-product 9.5 (PGP 9.5) (used as a pan-neuronal marker) and pituitary adenylate cyclase-activating polypeptide (PACAP) under physiological conditions (C) and after aspirin supplementation (ASA). The pictures (**A3**–**F3**) have been created by digital superimposition of two color channels, and colocalizations of both antigens in the studied neuron bodies were indicated with arrows. Scale bar size: 50 µm.

**Table 1 ijms-24-09871-t001:** Specification of immunoreagents.

**Primary Antibodies**				
**Antigen**	**Species**	**Dilution**	**Code**	**Supplier**
PGP 9.5	Mouse	1:1000	7863-2004	BioRad, Hercules, CA, USA
VIP	Rabbit	1:2000	ab22736	Abcam, UK
SP	Rabbit	1:1000	8450-0004	BioRad, Hercules, CA, USA
nNOS	Rabbit	1:3000	AB5380	Chemicon, USA
GAL	Rabbit	1:2000	4600-5004	Biogenesis, UK
PACAP	Guinea Pig	1:3000	T-5039	Peninsula, San Carlos, CA, USA
CART	Rabbit	1:16,000	H-003-61	Phoenix Pharmaceuticals, Inc., Burlingame, CA, USA
**Secondary Antibodies**				
**Reagents**		**Dilution**	**Code**	**Supplier**
Alexa Fluor 488 donkey anti-mouse IgG		1:1000	A21202	ThermoFisher Scientific, Waltham, MA, USA
Alexa Fluor 546 donkey anti-rabbit IgG		1:1000	A11010	ThermoFisher Scientific, Waltham, MA, USA
Alexa Fluor 546 donkey anti-guinea pig IgG		1:1000	A11074	ThermoFisher Scientific, Waltham, MA, USA

## Data Availability

All relevant data are contained within the manuscript.

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
