# Peer review of "Acetylsalicylic Acid Supplementation Affects the Neurochemical Phenotyping of Porcine Duodenal Neurons"

_ijms, 2023, doi:10.3390/ijms24129871_

Round 1
Reviewer 1 Report
The manuscript by Brzozowska and Całka deals with an up-to-date and unresolved problem, i.e. aspirin and its influence on the enteric nervous system. The results are elegantly presented, yet the manuscript is poorly (?wrongly) discussed.
Introduction:
It would be worth adding at the very beginning that: 1) low doses of ASA (typically 75 to 81 mg/day I humans) are sufficient to irreversibly acetylate serine 530 of COX-1, and this effect inhibits platelet generation of thromboxane A2, resulting in an antithrombotic effect; 2) while intermediate doses (650 mg to 4 g/day in humans) inhibit COX-1 and COX-2, blocking prostaglandin production, thus having analgesic and antipyretic effects. And in this context, the authors should clearly introduce their model and their aims, bearing in mind the pharmacology of ASA.
Besides, such information „porcine duodenum, which is particularly vulnerable to the adverse effects of acetylsalicylic acid” needs citation.
And at the same, citations such as: Palus, K.; Bulc, M.; Całka, J. Effect of Acrylamide Supplementation on the CART-, VAChT-, and nNOS-Immunoreactive Nervous 373 Structures in the Porcine Stomach. Animals (Basel). 2020, 10(4):555, should not be placed here - there are enough other citations in relation to the influence of NSAID on the ENS (but even worse, here this citations is given to describe structures of the ENS).
Methods:
„The preparations were subjected to double immunofluorescence staining to visualize the chemical coding of the enteric nervous system neurons. In the next step of the study,…” The ending and the beginning of the next paragraph should be rephrased.
„pre-absorption tests by incubating 321 tissues with an antibody that had been pre-absorbed with synthetic antigen (25 μg of ad‐ 322 equate antigen per 1 ml of the corresponding antibody in a working dilution)” - exact antigens and vendors should be added.
It would be worth adding how many sections per animal were prepared as well.
Discussion:
At first, the authors should refer to their model and pharmacological action of ASA (as we cannot find such information elsewhere). Why such dose, timing of the experiment and animals were used?
Authors should also comments on the fact that only duodenal and not gastric changes within ENS were analyzed. Besides, why this particular fragment of the intestine was used: „After the cessation of vital functions, tissues (3 cm duodenal fragments approximately 10 cm from the pylorus)” - previously (Rząp et al. Neurochemical Plasticity of nNOS-, VIP- and CART-Immunoreactive Neurons Following Prolonged Acetylsalicylic Acid Supplementation in the Porcine Jejunum), the authors wrote: „Fragments of jejunum (2 cm long) located approximately 45 cm from the pylorus were collected”?
And formostly, in majority, the results are discussed in the context of inflammation (???) but NOT in the context of the pharmacological action of ASA, i.e. prostaglandin-related. And how such changes within the ENS could affect the function of the gut?
Conclusions:
„It was also shown that nerve intramural cells show plasticity as a 353 result of pro-inflammatory factors. The analysis of changes in chemical coding within the 354 enteric nervous system will help to improve methods of preventing the development of 355 drug-induced inflammation in both humans and animals, as well as in understanding the 356 exact functions of selected neurochemicals.” - please rephrase, enigmatic. This conclusions are not supported by the presented results.
References:
This section should be reevaluated so only related papers are referenced to. Moreover, important references are lacking.
Minor editing of English language is required.
Author Response
Editor-in-Chief
International Journal of Molecular Sciences
You will find included corrected version of our manuscript entitled “Acetylsalicylic acid supplementation affects the neurochemical phenotyping of porcine duodenal neurons” – Marta Brzozowska, Jarosław Całka.
We appreciate the thorough review. All text improvements of our manuscript have been done in blue font.
Here are correction and comments from the reviewers:
- Reviewer 1
Introduction:
It would be worth adding at the very beginning that: 1) low doses of ASA (typically 75 to 81 mg/day I humans) are sufficient to irreversibly acetylate serine 530 of COX-1, and this effect inhibits platelet generation of thromboxane A2, resulting in an antithrombotic effect; 2) while intermediate doses (650 mg to 4 g/day in humans) inhibit COX-1 and COX-2, blocking prostaglandin production, thus having analgesic and antipyretic effects. And in this context, the authors should clearly introduce their model and their aims, bearing in mind the pharmacology of ASA.
Authors response:
Thank you for your valuable comment. The necessary information has been added.
Besides, such information „porcine duodenum, which is particularly vulnerable to the adverse effects of acetylsalicylic acid” needs citation.
Authors response:
Thank you for your comment. The authors have added the appropriate citation.
And at the same, citations such as: Palus, K.; Bulc, M.; Całka, J. Effect of Acrylamide Supplementation on the CART-, VAChT-, and nNOS-Immunoreactive Nervous 373 Structures in the Porcine Stomach. Animals (Basel). 2020, 10(4):555, should not be placed here - there are enough other citations in relation to the influence of NSAID on the ENS (but even worse, here this citations is given to describe structures of the ENS).
Authors response:
Thank you for your comment. The necessary changes have been made.
Methods:
„The preparations were subjected to double immunofluorescence staining to visualize the chemical coding of the enteric nervous system neurons. In the next step of the study,…” The ending and the beginning of the next paragraph should be rephrased.
Authors response:
Thank you for your valuable comment. The authors edited this part of the manuscript.
„pre-absorption tests by incubating 321 tissues with an antibody that had been pre-absorbed with synthetic antigen (25 μg of ad‐ 322 equate antigen per 1 ml of the corresponding antibody in a working dilution)” - exact antigens and vendors should be added. It would be worth adding how many sections per animal were prepared as well.
Authors response:
Thank you for your valuable comment. The authors added the necessary information.
Discussion:
At first, the authors should refer to their model and pharmacological action of ASA (as we cannot find such information elsewhere). Why such dose, timing of the experiment and animals were used?
Authors response:
Thank you for your valuable comment.
In the presented study, a domestic pig was used as a research model, due to the high probability of the human digestive system. The dose of aspirin used, as well as the duration of ASA supplementation, was aimed at determining how long-term (28 days) administration of high doses of aspirin (used in analgesic therapy) changes the phenotype of duodenal enteric neurons.
The authors added the necessary information.
Authors should also comments on the fact that only duodenal and not gastric changes within ENS were analyzed. Besides, why this particular fragment of the intestine was used: „After the cessation of vital functions, tissues (3 cm duodenal fragments approximately 10 cm from the pylorus)” - previously (Rząp et al. Neurochemical Plasticity of nNOS-, VIP- and CART-Immunoreactive Neurons Following Prolonged Acetylsalicylic Acid Supplementation in the Porcine Jejunum), the authors wrote: „Fragments of jejunum (2 cm long) located approximately 45 cm from the pylorus were collected”?
Authors response:
Thank you for your valuable comment. The authors added the necessary information.
The presented research is part of a grant aimed at determining how aspirin, naproxen and indomethacin supplementation affects the chemical coding of enteric neurons in the pig digestive tract. The initial sections of the gastrointestinal tract (stomach and duodenum) are particularly exposed to the harmful effects of NSAIDs.
Stomach results will be published soon.
The authors collected tissues located approximately 10 cm from the pylorus. This is where the major duodenal papilla is located in the pig. This selection was intended to collect the most similar duodenal section from all animals.
And formostly, in majority, the results are discussed in the context of inflammation (???) but NOT in the context of the pharmacological action of ASA, i.e. prostaglandin-related. And how such changes within the ENS could affect the function of the gut?
Authors response:
Thank you for your valuable comment. Relevant information has been added in the Discussion section
Conclusions:
„It was also shown that nerve intramural cells show plasticity as a 353 result of pro-inflammatory factors. The analysis of changes in chemical coding within the 354 enteric nervous system will help to improve methods of preventing the development of 355 drug-induced inflammation in both humans and animals, as well as in understanding the 356 exact functions of selected neurochemicals.” - please rephrase, enigmatic. This conclusions are not supported by the presented results.
Authors response:
Thank you for your valuable comment. This paragraph has been changed.
References:
This section should be reevaluated so only related papers are referenced to. Moreover, important references are lacking.
Authors response:
Thank you for your comment. Relevant references have been added.

Reviewer 2 Report
This manuscript by Brzozowska et al. describes Acetylsalicylic acid supplementation affects the neurochemical phenotyping of porcine duodenal neurons. The work is well planned, results and discussion are well-written in this manuscript and falls in the scope of the journal and special issue However, some suggestions will improve the quality of this manuscript and may be considered for publication in IJMS after below comments.
1. What was the criteria for dose 100 mg/kg in pigs (it’s high dose)?
2. Did you observe the progression of other side effect like bleeding etc.
3. Acetylcholine is one of the neurotransmitters present in gastrointestinal neurons, did authors observe its level in this study?
4. Authors should provide the quantification data for these neurotransmitters.
5. In discussion authors need to discuss and include the link between theses neurotransmitter observed in the study.
6. This study will more laudable if author include the role of neuro-glia interactions as glial cells has primary role in inflammation
No Comments
Author Response
Editor-in-Chief
International Journal of Molecular Sciences
You will find included corrected version of our manuscript entitled “Acetylsalicylic acid supplementation affects the neurochemical phenotyping of porcine duodenal neurons” – Marta Brzozowska, Jarosław Całka.
We appreciate the thorough review. All text improvements of our manuscript have been done in blue font.
Here are correction and comments from the reviewers:
- Reviewer 2
- What was the criteria for dose 100 mg/kg in pigs (it’s high dose)?
Authors response:
Thank you for your comment. In the presented manuscript, we show data describing the effect of long-term use of high doses of aspirin on the phenotype of porcine duodenal neurons. Such high doses are often used in humans in the treatment of inflammation, e.g. rheumatoid disease. Earlier studies also used this dosage:
- Palus K, Integra J. The Influence of Prolonged Acetylsalicylic Acid Supplementation-Induced Gastritis on the Neurochemistry of the Sympathetic Neurons Supplying Prepyloric Region of the Porcine Stomach. PLOS One. 2015 Nov 25;10(11):e0143661.
- Rytel L, Calka J. Acetylsalicylic acid-induced changes in the chemical coding of extrinsic sensory neurons supplying the prepyloric area of the porcine stomach. Neurosci Lett. 2016 Mar 23;617:218-24.
- Did you observe the progression of other side effect like bleeding etc.
Authors response:
Thank you for your valuable comment. This information has been added in the Results section.
- Acetylcholine is one of the neurotransmitters present in gastrointestinal neurons, did authors observe its level in this study?
Authors response:
Thank you for your valuable comment. Acetylcholine is a neurotransmitter widely distributed in the enteric nervous system. Nevertheless, the present study aimed to determine the expression of neurosubstances involved in inflammatory processes in the porcine duodenum after ASA administration.
Immunoreactivity of enteric neurons against acetylcholine after aspirin supplementation is a research topic that should be explored in the future.
- Authors should provide the quantification data for these neurotransmitters.
Authors answer:
Thank you for your comment.
The table presents percentages of immunoreactive neurons in the wall of the duodenum in the control (C group) and experimental animals (ASA group). Results from each animal, mean and exact p-value are shown. In the manuscript, data are presented as mean ± SEM.
|
Active neuronal substances studied during the experiment |
Type of group |
Number of animal |
Myenteric plexus |
Outer submucous plexus |
Inner submucous plexus |
|
|
VIP |
C group |
1 |
11.94% |
13.08% |
19.54% |
|
|
2 |
12.28% |
12.13% |
15.81% |
|||
|
3 |
12.33% |
12.99% |
17.28% |
|||
|
4 |
11.03% |
16.93% |
17.90% |
|||
|
Mean |
11.90% |
13.78% |
17.63% |
|||
|
ASA group |
1 |
12.34% |
21.34% |
22.25% |
||
|
2 |
14.45% |
20.47% |
22.75% |
|||
|
3 |
14.40% |
20.68% |
19.84% |
|||
|
4 |
13.94% |
20.08% |
22.31% |
|||
|
Mean |
13.78% |
20.64% |
21.78% |
|||
|
Value of p |
p=0.004736 |
p=0.0005681 |
p=0.003349 |
|||
|
nNOS |
C group |
1 |
18.08% |
7.87% |
4.50% |
|
|
2 |
18.34% |
9.27% |
5.98% |
|||
|
3 |
19.26% |
7.16% |
7.72% |
|||
|
4 |
18.30% |
9.31% |
7.34% |
|||
|
Mean |
18.50% |
8.70% |
6.39% |
|||
|
ASA group |
1 |
20.12% |
11.29% |
10.63% |
||
|
2 |
21.03% |
9.09% |
10.43% |
|||
|
3 |
20.16% |
11.44% |
10.78% |
|||
|
4 |
19.57% |
12.03% |
9.81% |
|||
|
Mean |
20.22% |
10.96% |
10.41% |
|||
|
Value of p |
p=0.0018639 |
p=0.006808 |
p=0.01157 |
|||
|
SP |
C group |
1 |
10.55% |
24.31% |
21.30% |
|
|
2 |
7.86% |
18.60% |
20.92% |
|||
|
3 |
7.88% |
18.63% |
20.87% |
|||
|
4 |
8.65% |
19.32% |
20.12% |
|||
|
Mean |
8.74% |
20.22% |
20.80% |
|||
|
ASA group |
1 |
15.71% |
22.99% |
22.55% |
||
|
2 |
14.79% |
25.79% |
22.66% |
|||
|
3 |
14.73% |
25.82% |
22.91% |
|||
|
4 |
14.68% |
25.95% |
22.11% |
|||
|
Mean |
14.98% |
25.14% |
22.56% |
|||
|
Value of p |
p=0.001919 |
p=0.013852 |
p=0.00015488 |
|
||
|
GAL |
C group |
1 |
1.18% |
19.96% |
35.63% |
|
|
2 |
1.39% |
16.00% |
28.24% |
|
||
|
3 |
1.17% |
19.65% |
26.69% |
|
||
|
4 |
1.69% |
23.22% |
28.12% |
|
||
|
Average |
1.36% |
19.71% |
29.67% |
|
||
|
ASA group |
1 |
3.37% |
26.55% |
35.19% |
|
|
|
2 |
3.57% |
27.22% |
37.52% |
|
||
|
3 |
3.37% |
26.63% |
34.06% |
|
||
|
4 |
3.80% |
26.43% |
35.06% |
|
||
|
|
3.53% |
26.71% |
35.46% |
|
||
|
Value of p |
p=0.000628 |
p=0.0017305 |
p=0.02471 |
|
||
|
PACAP |
C group |
1 |
10.29% |
11.22% |
11.81% |
|
|
2 |
9.33% |
9.87% |
13.41% |
|
||
|
3 |
8.81% |
10.12% |
12.90% |
|
||
|
4 |
10.33% |
10.82% |
13.17% |
|
||
|
Average |
9.69% |
10.51% |
12.82% |
|
||
|
ASA group |
1 |
14.17% |
14.79% |
16.14% |
|
|
|
2 |
13.86% |
16.10% |
17.82% |
|
||
|
3 |
13.89% |
15.58% |
18.06% |
|
||
|
4 |
14.17% |
15.97% |
17.76% |
|
||
|
Average |
14.02% |
15.61% |
17.45% |
|
||
|
Value of p |
p=0.00006686 |
p=0.000009 |
p=0.0001024 |
|
||
|
CART |
C group |
1 |
14.40% |
16.10% |
10.30% |
|
|
2 |
12.92% |
15.84% |
10.91% |
|
||
|
3 |
14.09% |
16.11% |
10.35% |
|
||
|
4 |
13.01% |
15.46% |
10.65% |
|
||
|
Average |
13.61% |
15.88% |
10.55% |
|
||
|
ASA group |
1 |
15.19% |
19.76% |
14.20% |
|
|
|
2 |
15.54% |
20.36% |
12.30% |
|
||
|
3 |
15.71% |
19.96% |
12.87% |
|
||
|
4 |
15.23% |
20.92% |
14.80% |
|
||
|
Average |
15.42% |
20.25% |
13.54% |
|
||
|
Value of p |
p=0.009309 |
p=0.0002442 |
p=0.002364 |
|
||
- In discussion authors need to discuss and include the link between theses neurotransmitter observed in the study.
Authors response:
Thank you for your valuable comment. Relevant information has been added in the Discussion section.
- This study will more laudable if author include the role of neuro-glia interactions as glial cells has primary role in inflammation
Authors response:
Thank you for your comment. Relevant information has been added in the Discussion section.

Round 2
Reviewer 1 Report
The Authors did not sufficiently answered the previous remarks.
Why Authors did not cited their own work? - Rząp et al. Neurochemical Plasticity of nNOS-, VIP- and CART-Immunoreactive Neurons Following Prolonged Acetylsalicylic Acid Supplementation in the Porcine Jejunum
The manuscript should be checked by an English specialist.
Author Response
Thank you for your valuable comments. The manuscript was revised according to previous reviewer's directions (discussion section).
The missing citation has been added (number 7) and the English has been corrected.
Reviewer 2 Report
Authors revised manuscript as per suggestions. Current version can be considered for publication.
Author Response
Thank you for the thorough review.